# One-pot Golden Gate Assembly of an avian infectious bronchitis virus reverse genetics system

Katharina Bilotti[1], Sarah Keep[2], Andrew P. Sikkema[1], John M. Pryor[1], James Kirk[2], Katalin Foldes[2], Nicole Doyle[2], Ge Wu[2], Graham Freimanis[2], Giulia Dowgier[3], Oluwapelumi Adeyemi[2], S. Kasra Tabatabaei[1], Gregory J. S. Lohman[1]*, Erica Bickerton[2]*

**1** New England Biolabs, Ipswich, Massachusetts, United States of America, **2** The Pirbright Institute, Woking, United Kingdom, **3** The Francis Crick Institute, London, United Kingdom

☯ These authors contributed equally to this work.

\* erica.bickerton@pirbright.ac.uk (EB); lohman@neb.com (GJSL)

## Abstract

Avian infectious bronchitis is an acute respiratory disease of poultry of particular concern for global food security. Investigation of infectious bronchitis virus (IBV), the causative agent of avian infectious bronchitis, via reverse genetics enables deeper understanding of virus biology and a rapid response to emerging variants. Classic methods of reverse genetics for IBV can be time consuming, rely on recombination for the introduction of mutations, and, depending on the system, can be subject to genome instability and unreliable success rates. In this study, we have applied data-optimized Golden Gate Assembly design to create a rapidly executable, flexible, and faithful reverse genetics system for IBV. The IBV genome was divided into 12 fragments at high-fidelity fusion site breakpoints. All fragments were synthetically produced and propagated in *E. coli* plasmids, amenable to standard molecular biology techniques for DNA manipulation. The assembly can be carried out in a single reaction, with the products used directly in subsequent viral rescue steps. We demonstrate the use of this system for generation of point mutants and gene replacements. This Golden Gate Assembly-based reverse genetics system will enable rapid response to emerging variants of IBV, particularly important to vaccine development for controlling spread within poultry populations.

## Introduction

Coronaviruses are a family of viruses that infect birds and mammals and that share a large (approximately 30kb) positive sense RNA genome structure, capped at the 5′ end and polyadenylated at the 3′ end. There are four genera, *Alpha-*, *Beta-*, *Gamma-* and *Deltacoronavirus*, with members of each genus posing threats to human and animal health. Genome organization is shared between the genera with a large replicase gene composed of two open reading frames (ORF) that code for the polyproteins pp1a and pp1ab. The majority of the rest of the

**Data Availability Statement:** The genomic sequence for D388-WT has been uploaded to GenBank under accession number OR813926. All

other relevant data are within the manuscript and its Supporting Information files.

**Funding:** When performing this research and drafting this manuscript, KB, APS, JMP, SKT, and GJSL were employees of New England Biolabs, Ipswich, MA (neb.com). The funder provided support in the form of salaries for KB, APS, JMP, SKT, and GJSL but did not have any additional role in the study design, data collection and analysis, decision to publish, or preparation of the manuscript. Funding for this project was provided by the Biotechnology and Biological Sciences Research Council (BBSRC; https://www.ukri.org/councils/bbsrc/), grants BBS/E/I/00001424, BBS/E/I/00007031, BBS/E/I/00007034, BBS/E/I/00007035, BBS/E/I/00007037, BBS/E/I/00007038, BBS/E/I/00007039, BBS/E/PI/230001B, BBS/E/PI/230002A, BBS/E/PI/23NB0004 and BBS/E/PI/23NB0003 to EB, and by strategic funding to The Pirbright Institute. The funders did not play any role in the study design, data collection and analysis, decision to publish, or preparation of the manuscript.

**Competing interests:** I have read the journal's policy and the authors of the manuscript have the following competing interests: When performing this research and drafting this manuscript, KB, APS, JMP, SKT, and GJSL were employees of New England Biolabs, a manufacturer and vendor of molecular biology reagents including DNA ligases and Type IIS restriction enzymes. New England Biolabs funded the work and paid the salaries of these authors. This does not alter our adherence to PLOS ONE policies on sharing data and materials. A patent has previously been filed by The Pirbright Institute to protect the intellectual property of the work surrounding the mutations in nsp 10 and nsp 14 (Patent name: Coronavirus, Number EP3172319B1, Authors: Erica Bickerton, Sarah Keep, and Paul Britton). This does not alter our adherence to PLOS ONE policies on sharing data and materials.

genome contains ORFs coding for the structural proteins spike (S), envelope (E), membrane (M) and nucleocapsid (N), with a varying number of accessory genes interspersed. Flanking the genome are two untranslated regions, denoted the 5′ and 3′ UTR.

Avian infectious bronchitis virus (IBV), a *gammacoronavirus* and the first coronavirus identified [1–3], is the etiological agent of infectious bronchitis, an acute highly contagious predominantly respiratory disease of domestic fowl. IBV has a global distribution with many circulating and co-circulating strains that are classified by both serotype and genotype [4–6]. Many IBV strains inflict classical respiratory disease characterized by snicking, tracheal rales, watery eyes, nasal discharge, lethargy, reduced weight gain and drops in egg production and quality. Other IBV strains, for example QX, which emerged in the mid 2000's in China [7] and subsequently spread to Europe as D388(QX) [8], are deemed nephropathogenic and are characterized by additional symptoms including nephritis which can lead to high mortality and false layer syndrome. Many strains of IBV also render chickens susceptible to secondary bacterial infections, increasing mortality and spread of disease among agricultural populations. While vaccination is routinely practiced, cross protection between IBV strains is poor. Consequently, young chicks typically receive multiple live attenuated vaccines in the first few weeks of life [5]. Currently these vaccines are generated through the time-consuming process of serial passage of a virulent isolate through embryonated hens' eggs, typically 80 to 100 times [9, 10]. A fine balance must be achieved between attenuation and retention of immunogenicity and, as the molecular mechanism of attenuation is unknown, vaccines present a risk of reversion [11–13]. Alongside recombination in the field and/or the accumulation of genetic mutations, this potential leads to constantly emerging new variants and strains despite vaccination. Despite nearly a century of research, IBV remains a major concern to both the global poultry industry and global food security.

While IBV is predominantly of global concern due to risks to food security, other coronaviruses present risks for both domestic animals and human health. In the 20th and 21st centuries, zoonotic transfer events have led to the emergence of novel pathogens and diseases with health and economic impacts, both veterinary and human. The recently emerged swine acute diarrhoea syndrome coronavirus (SADS-CoV) causes acute diarrhoea in neonatal piglets and has resulted in significant economic losses in the pork industry in both Asia and the United States of America [14, 15]. Although human infection has not been reported and farm workers in the original outbreak showed no seroconversion (suggesting no infection) [16], SADS-CoV is considered a risk to humans due to the viruses' ability to replicate in several human cell lines [17]. Likewise, the porcine delta coronavirus (PDCoV) that emerged in 2012, which also causes gastrointestinal symptoms in piglets, is considered a risk for human infection [18–20]. Zoonotic transfer events resulting in severe disease are well documented for the human coronaviruses, including severe acute respiratory syndrome coronavirus (SARS-CoV) and Middle East respiratory syndrome coronavirus (MERS-CoV) which emerged in 2002 and 2012 respectively [21, 22]. Most recently, SARS-CoV-2 emerged in late 2019 with the resulting pandemic estimated to have resulted in over 6 million deaths worldwide as well as severe economic impact across the globe [23]. Understanding the disease biology of and developing robust vaccines against this important group of pathogens is therefore paramount for the development of effective treatments and control measures, including biosecurity protocols and vaccines, to protect both public health and global food security.

The study of coronavirus biology, including that of IBV, has been greatly enhanced by reverse genetics which allows for specific modification of the coronavirus genome, commonly in the search for antiviral treatment and vaccine development [2, 4, 24–26]. The first method of modifying the coronavirus genome was described in the 1990s and was based on RNA recombination. It was originally based on Murine Hepatitis Virus (MHV) using temperature

sensitive lesions within the N gene. While this was a significant advancement, the technology was limited in application and did not easily allow for modification of all areas of the genome; it was therefore was not widely adopted for all coronaviruses. Classic reverse genetics systems enabling the modification of a full length cDNA copy of the coronavirus genome were first developed in the early 2000's based on RNA recombination, *in vitro* ligation using pre-existing or uniquely engineered restriction sites, or assembly and propagation of a cDNA copy of the coronavirus genome in a Bacterial Artificial chromosome (BAC) or Vaccinia Virus Vector [27–29]. More recently, several other systems have been established including the use of yeast [30], overlapping PCR fragments following a circular polymerase extension reaction (CPER) methodology [31], Lambda Red-based recombination [32] and Type IIS restriction endonucleases [33]. Each system has limitations, including time, genome stability, error rate, and/or ease of manipulation, with some systems more suitable for specific coronaviruses than others. For example, RNA recombination does not permit the modification of ORF 1a or 1ab in coronaviruses [34] and *in vitro* ligation methods typically rely on *in vitro* transcription followed by the electroporation of the full-length RNA transcript into a susceptible cell line, which is a particular difficulty if using primary cells for the subsequent recovery of infectious recombinant virus. Sequence stability and integrity is an additional concern—for example, the IBV genomic cDNA, when propagated via BAC, is unstable in *Escherichia coli (E. coli)*. There are also logistical considerations, such as the enhanced biosafety measures required for working with vaccinia virus and the additional laboratory facilities separate from cell culture suites that are required for propagation of yeast. Reverse genetics for IBV specifically has historically been confined to RNA recombination [35] or to the vaccinia-based system [36–38] which utilizes homologous recombination for the modification of the IBV cDNA. Whilst the error rate of the latter system is very low and the system provides the necessary quantities of cDNA for subsequent recovery of infectious virus in primary chicken kidney (CK) cells, the process can be slow and cumbersome and is not well suited to rapid reaction to emerging IBV strains and/or variants. The development of a reverse genetics system that 1) requires minimal *in vitro* amplification or *in vivo* passage of the fully assembled genome, thereby reducing the potential for sequence error, 2) provides the necessary quantities of input nucleic acid for efficient recovery in primary cells, and 3) uses equipment and reagents that are routinely used in molecular biology laboratories is therefore highly desirable. Such a system would not only be beneficial for IBV but would also be applicable to the wider coronavirus field.

Golden Gate Assembly (GGA) is widely used in synthetic biology to create systems for studying gene networks and metabolic pathways, and was recently applied to develop a new reverse genetics system for another coronavirus, the betacoronavirus SARS-CoV-2 [39]. GGA permits the seamless assembly of large DNA constructs from multiple smaller fragments [40]. GGA relies on the action of Type IIS restriction enzymes, which cleave outside of their recognition sequence to create short overhangs, typically three or four bases in length, with no sequence constraint. The assembly order of DNA fragments can be specified using sets of overhang pairs which are joined by a DNA ligase. GGA uses the restriction enzyme and ligase in a single step, cycled reaction to permit the simultaneous joining of multiple fragments in a defined order in one operation. GGA relies on accurate ligation to maximize the yield of properly assembled constructs, as mismatch ligation can lead to constructs with mis-ordered, duplicated, or missing fragments. Classic application of GGA typically supports assemblies of 2–8 parts at once, with larger assemblies built over successive rounds of assembly. Recent work has used comprehensive profiles of DNA ligase fidelity to identify overhang combinations with high mismatch potential and avoid them when selecting fusion sites [41, 42]. This has significantly expanded the capacity of GGA, with data-optimized assembly design (DAD) enabling the routine assembly of 25–35 fragments in a single reaction with high yields of the desired

assembly with a low error rate [43, 44]. The expanded one-pot capacity provided by DAD has recently enabled the application of this methodology to simplify viral genome construction, permitting rescue of the T7 bacteriophage genome from 12–52 fragments in a two day protocol [45]. The ability to divide the genome into many parts provides flexibility in the placement of fusion sites, allowing open reading frames (ORFs) or other regions of interest to be isolated on individual fragments and permitting manipulation of focused regions with re-use of parts in regions that do not need to be altered. Further, the part size can be reduced such that the DNA can be produced synthetically and/or cloned into carrier plasmids, enabling simple manipulation of the sequence *in silico* or through classic *E. coli*-compatible molecular biology techniques.

In the current publication, we describe the application of DAD to the development of a GGA-based reverse genetics system for the D388 strain of avian IBV (D388-WT). This system permits the rapid and flexible assembly of complete viral cDNA from 12 parts held in plasmids, rationally designed to isolate many ORFs in fragments that are readily propagated and manipulated in traditional *E. coli* plasmid systems. The viral cDNA is assembled in one round in high yield and accuracy, permitting direct rescue in primary cells without intermediate passage and screening in a secondary organism. The resulting recombinant IBV (rIBV), denoted D388-GGA, was characterized and compared to D388-WT. Replication was comparable *in vivo* and comparable titers were achieved *in ovo* at 24 hours post infection (hpi). D388-GGA also exhibited a comparable pathogenic phenotype *in vivo* demonstrating that the reverse genetics system developed is a powerful tool for the study of pathogenic and immunogenic determinants. Further, the design enables the simplified introduction of specific mutations via site directed mutagenesis in component plasmids and enables genetic knockouts and gene replacements through substitution of individual parts, as demonstrated by the generation of three further rIBVs. The assembled cDNA is compatible with classic methods for viral rescue and directly usable in transfection without any need for passage in *E. coli* or vaccinia virus. The reverse genetics system presented will enable easier and more rapid investigation of virus biology, replication, and virulence. It further presents a facile platform for accelerating vaccine research and development for avian IBV, with advantages of speed, efficiency, fidelity, and safety compared to other systems utilizing Type IIS restricted parts, yeast assembly, or passage through *E. coli*.

## Results

### Design and assembly of a GGA-based reverse genetics system for avian IBV

A 28 kb consensus sequence for the IBV strain D388 was established using Illumina Sequencing and an in-house assembly pipeline [46] with the 5′ end of the genome determined using 5′ end rapid amplification of cDNA ends (RACE) (GenBank accession number OR813926). This sequence was used as the basis for the *in silico* design of the GGA assembly system, with additional features incorporated to allow for both efficient assembly and virus recovery (S1 File). First, two silent mutations were introduced into the genome to remove native BsaI recognition sites (T735A and T891A, S1 Table). BsaI was chosen as the enzyme to be used in the assembly as the D388 genome contained only two native BsaI restriction sites, both within the ORF for non-structural protein (nsp) 2, permitting confident removal without risking major gene regulation effects that could be caused by modifying intergenic regions. These mutations subsequently act as marker mutations to enable differentiation between virus derived from successful assembly and wild type virus contamination. Additionally, a T7 promoter was added upstream of the 5′ UTR in the cDNA construct to allow high yield conversion to the RNA genome [36]. Putting the genome cDNA under a T7

promoter is advantageous over a cytomegalovirus (CMV) promoter as it avoids the possibility of splicing during nuclear expression and ensures that no genomic IBV RNA can be produced from transfected cells unless the T7 RNA polymerase gene is simultaneously introduced into the cells. Downstream of the 3′ UTR, an encoded poly A tail, a T7 terminator sequence and a hepatitis delta ribozyme sequence were added; the latter permits self-cleavage of the transcript *in vitro* to leave a uniform 3′ end [36]. Finally, the assemblies were designed to form a circular construct to remove uncertainty about the end structure and provide protection against cellular nucleases.

With the full circular target sequence for the assembly designed, we applied our previously described SplitSet tool (ligasefidelity.neb.com) to select high fidelity junction sets within the sequence [43, 44]. The sequence was rationally divided into 12 fragments (Fig 1, S1 Fig and S2 Table) generally restricting the fusion sites to the regions between ORFs, with an additional fusion site placed at each end of the genome as the circularization fusion site. This rational breakpoint strategy ensured that sequences of interest were grouped together or isolated into a single fragment to allow for easy manipulation and interrogation of individual genes and proteins. As an example of the strategy considerations, the S gene encoding the spike protein, which is essential to study for antiviral and vaccine development, is contained in its entirety on fragment D388-F10 (Fig 1). While it is feasible to divide the genome further, 12 fragment assemblies with DAD-selected fusion sites have been shown to assemble with very high fidelity and efficiency, while permitting flexible junction placement and the ability to add additional junctions as needed [44]. The design strategy and the set of overhangs chosen for this assembly were predicted to assemble with a 97% fidelity (S2 Fig), effectively expected to be near perfect assembly accuracy, and previous work has shown high yields of full-length product are possible with 12 fragment assemblies giving rise to a very high number of CFUs or PFUs in other systems [44, 45]. This design results in an average fragment size of 2.5 kb, a convenient size for manipulation within standard *E. coli* plasmids, and small enough such that the parts were able to be produced as synthetic fragments. Assembly from plasmid parts resulted in primarily full-length DNA product (S3 Fig) that was used in virus recovery protocols without further purification.

## Recovery of recombinant IBV D388-GGA from transfection of assembled cDNA

Following cDNA assembly, the rescue of infectious recombinant IBV D388-GGA was carried out in primary chicken kidney (CK) cells using a previously published protocol [47]. Briefly, CK cells were infected with a recombinant fowlpox that expressed T7 RNA polymerase and subsequently transfected with D388-GGA cDNA and a plasmid expressing the D388 N protein, also under the control of a T7 promoter. The recovery of all infectious GGA-generated IBVs was tested in triplicate with the resulting cell lysate inoculated into embryonated hens' eggs for the amplification of newly generated rIBV. The presence of D388-GGA in the harvested allantoic fluid was confirmed by reverse transcriptase (RT)-PCR using primers targeting the D388 N gene and 3′UTR. All three replicates were positive for IBV derived RNA with one taken forward for the generation of stock virus, demonstrating that the assembly method was compatible with the existing well-established method for rIBV recovery. Next Generation Sequencing of the stock virus confirmed that the sequence of the D388-GGA IBV genome matched the target design and no mutations were identified. The presence of the two intentionally introduced silent mutations to differentiate D388-GGA from wild type D388 (D388-WT) was also confirmed. Next generation sequencing generated 1099234 reads which covered 100% of the genome with an average coverage depth of 9487.86.

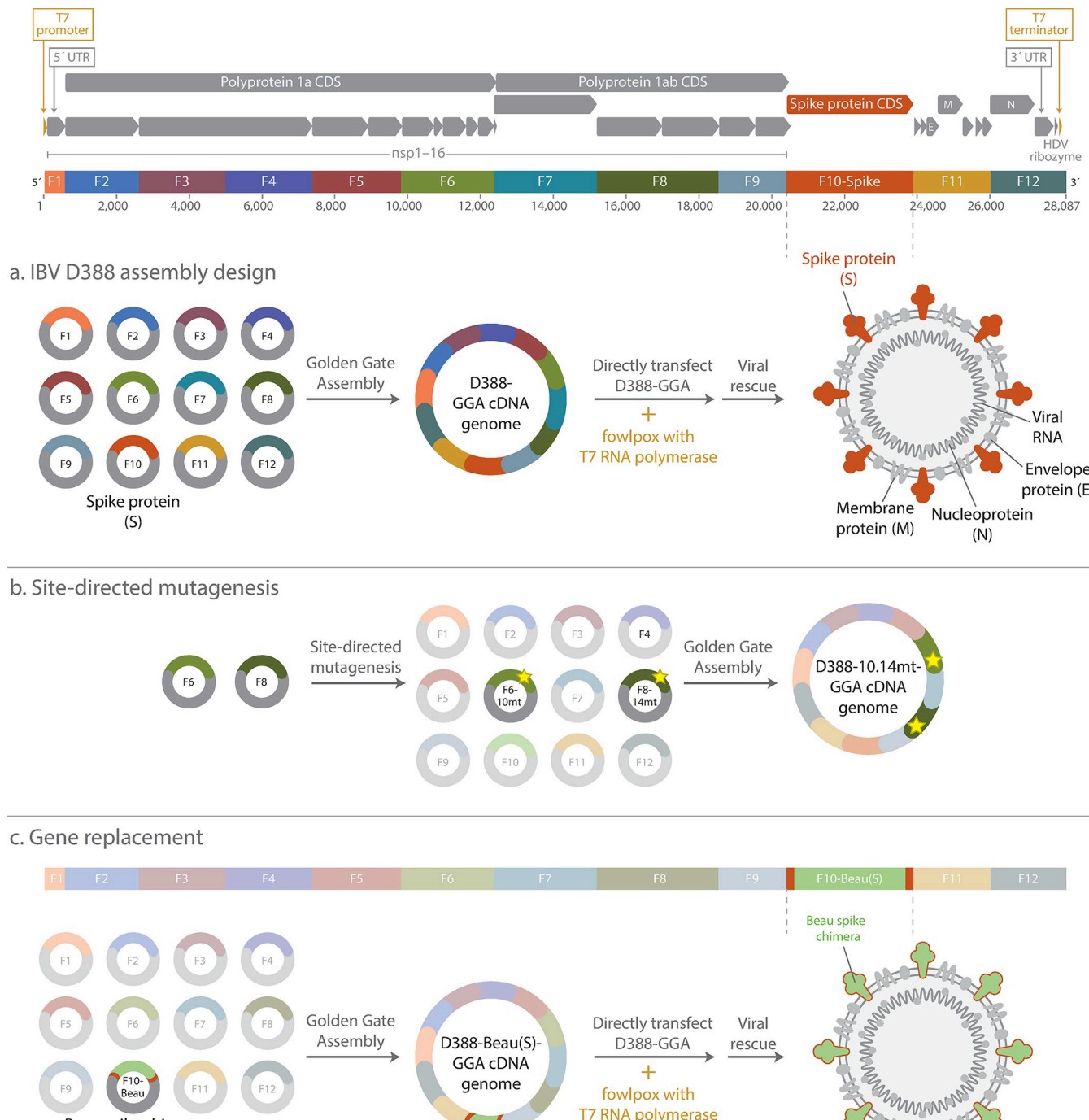

**Fig 1. Design and Golden Gate Assembly of an IBV D388 reverse genetics system.** The 28 kb IBV D388 genome was used as the basis for a GGA-based reverse genetics system. Several additional features, including a T7 promoter, T7 terminator, and HDV ribozyme sequence, were incorporated into the design to permit conversion of assembled cDNA into RNA. The resulting sequence was split into 12 fragments, restricting the break points to be between ORFs, and was designed to assemble in a circular construct. Notably, the entire S gene, containing the spike protein, was designed to be contained on a single fragment (D388-F10) for easy manipulation. **A** IBV D388 cDNA was assembled from plasmids containing fragments D388-F1 through D388-F12 and directly co-transfected with a fowlpox virus expressing T7 RNA polymerase before virus rescue. **B** IBV variants containing point mutants were generated using SDM on plasmids D388-F6 and D388-F8 before assembly. **C** The IBV D388 S gene was substituted with the Beau-R S gene to generate IBV containing the Beau-R spike protein.

## Characterization of D388-GGA

Replication kinetics of D338-GGA IBV were subsequently assessed *in ovo*, *in vitro* and within *ex vivo* tracheal organ cultures (TOCs) and compared to D388-WT (Fig 2). Infection *in ovo* resulted in equivalent titres between D388-GGA and D388-WT indicating comparable levels of replication (Fig 2A). The replication of IBV *in vivo* occurs primarily in ciliated epithelial cells that line the trachea and results in the cessation of cilia movement, referred to as ciliostasis [48, 49]. Tracheal ciliary activity is therefore used as a marker for IBV replication. In *ex vivo* TOCs ciliary activity was comparable between D388-GGA and D388-WT from 48 to 96 hours post infection (hpi) with both viruses able to cause ciliostasis. Ciliary activity was measured at both 37˚C and 41˚C as the respiratory tract exhibits a temperature gradient (Fig 2B). Lower temperatures (37˚C) are exhibited in the upper tract compared to the lower tract, which is closer to body temperature at 41˚C. Interestingly at 24 hpi at both 37˚C and 41˚C ciliary activity was higher in TOCs infected with D388-GGA in comparison to D388-WT suggesting a difference in replication in this assay. Differences in replication were also observed *in vitro* (Fig 2C). CK cells were inoculated with either D388-GGA, D388-WT or mock infected with media only and were stained with an antibody targeting dsRNA, a marker for IBV replication [50, 51] and imaged using confocal microscopy. Unlike that observed in D388-WT infected cells, no dsRNA could be detected in those infected with D388-GGA suggesting D388-GGA is

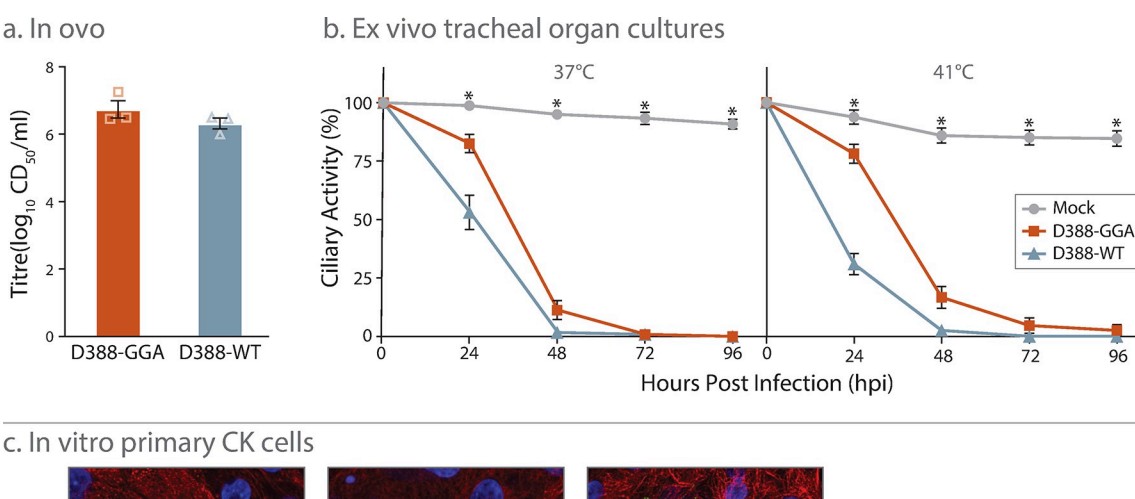

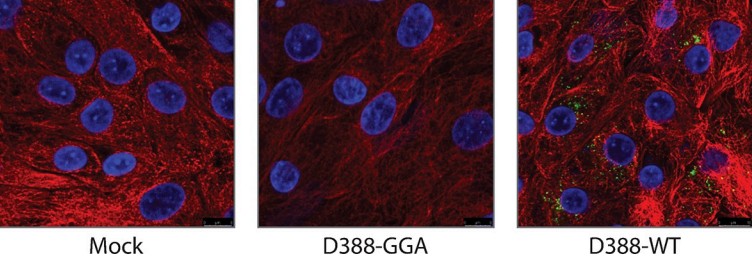

**Fig 2. Replication of D388-GGA is comparable to D388 WT in *ex vivo* TOCs and *in ovo*. A** 10 day old SPF eggs were inoculated with $10^3$ $CD_{50}$ of either D388-GGA or D388-WT. The quantity of infectious progeny in the allantoic fluid 24 hpi was determined via titration. **B** In replicates of 10 *ex vivo* TOCs were inoculated with $10^3$ $CD_{50}$ of either D388-GGA, D388-WT or mock infected with media. Infected TOCs were incubated at either 37 or 41˚C and the ciliary activity observed at 24 h intervals. For A and B, error bars represent standard error of the mean (SEM) of three independent experiments, and statistical differences are highlighted by * (p<0.05). For panel B, statistical differences highlighted for both 37 and 41˚C are between mock and both D388-GGA and D388-WT, 24 to 96 hpi and between D388-WT and D388-GGA at 24 hpi only. **C** Chicken Kidney cells were mock infected or infected with MOI > 1 of D388-GGA or D388-WT. Infected cells were fixed 24 h post infection and immunolabelled with mouse double stranded RNA (dsRNA) monoclonal antibody and rabbit anti-tubulin. Primary antibodies were visualized with Alexa Fluor 488 nm conjugated goat anti-mouse (green) and Alexa Fluor 568 nm conjugated goat anti-rabbit (red). Nuclei were labelled with DAPI (blue).

unable to establish infection in primary CK cells. This was further supported through additional assays including RT-PCR analysis of D388-GGA infected CK cells in which no IBV derived RNA could be detected, plaque assays in which no viral plaques could be observed, and via serial passaging in which no virus could be detected.

## D388-GGA exhibits a pathogenic phenotype *in vivo*

It is important, particularly for the study of pathogenic and immunogenic determinants within the IBV genome, that the molecular clone D388-GGA exhibits a pathogenic phenotype *in vivo*. To determine *in vivo* pathogenicity, SPF RIR chicks were inoculated with either D388-GGA, D388-WT or mock infected with phosphate buffered saline (Fig 3). The level of snicking (Fig 3A) and tracheal rales (Fig 3B) observed in chicks infected with D388-GGA from 1 to 7 days post infection (dpi) were comparable to those infected with D388-WT, with a modest lag seen in the D388-GGA infected chicks. The similar levels of snicking and rales demonstrate that infection with either D388-GGA or D388-WT results in similar clinical disease progression. Further, trachea were extracted from 5 randomly selected chicks 4 and 6 dpi and the ciliary activity assessed. At both 4 and 6 dpi, both D388-GGA and D388-WT compromised ciliary activity to similar levels. (Fig 3C). The observed reduction in tracheal ciliary activity indicates comparable *in vivo* viral replication. To further confirm the extent and distribution of *in vivo* replication, RNA was extracted from trachea, lung, kidney and eyelid tissues, which represent

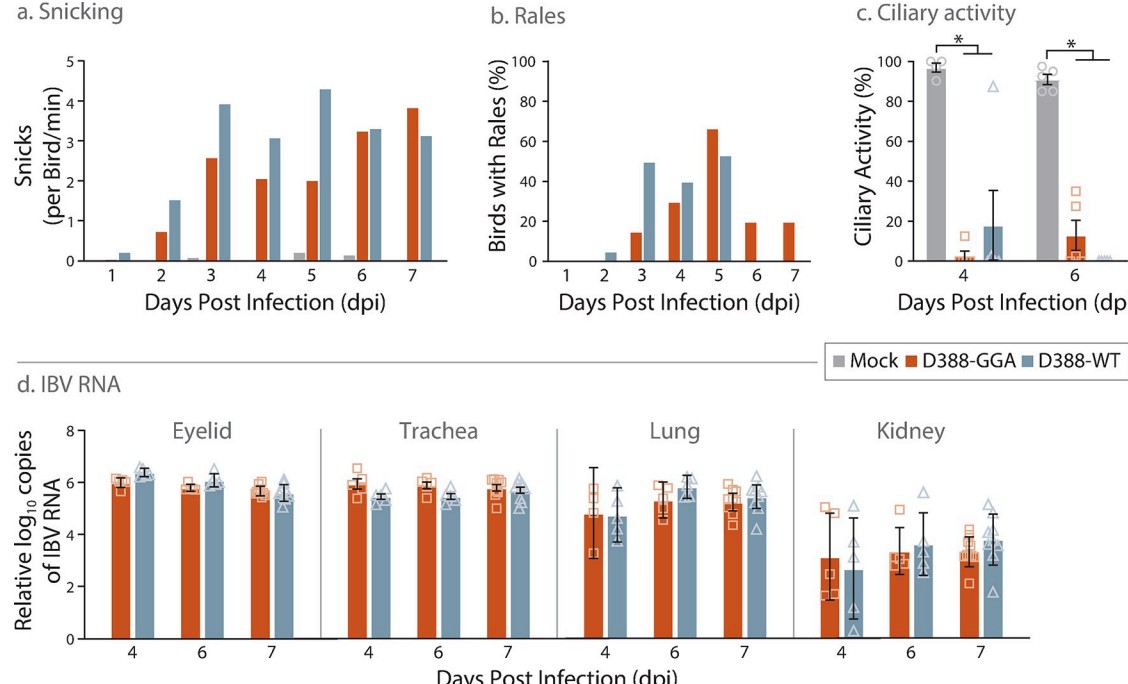

**Fig 3. D388-GGA exhibits a pathogenic phenotype *in vivo*.** Groups of 20 SPF RIR chickens at 7 days of age were inoculated via the intranasal and intraocular route with $10^{3.6}$ $CD_{50}$ either D388-GGA, D388-WT or mock infected with PBS. From 1 to 7 days post infection (A) the number of snicks per bird per min and (B) the percentage of birds exhibiting rales were calculated. On days 4 and 6 dpi the ciliary activity in trachea extracted from 5 randomly chosen birds was calculated. The average percentage of ten tracheal rings observed for each bird is presented. (D) RNA was extracted from tissues harvested from 5 randomly selected birds 4 and 6 dpi and 10 birds on 7 dpi. RNA was assessed by RT-qPCR for the quantity of IBV derived RNA using primers and probes specific for the 5′UTR. Each point represents a single bird. (C, D) Error bars represent SEM. Statistical differences highlighted by * ($p < 0.05$) were assessed using a two-way ANOVA with a Tukey test for post hoc analysis. For panels C and D no statistical differences were identified between D388-GGA and D388-WT.

the established known sites of IBV replication [49, 52], and the quantity of IBV-derived RNA was assessed using reverse transcription quantitative real time PCR (RT-qPCR) (Fig 3D). Equivalent quantities of IBV derived RNA were detected from eyelid, trachea, lung, and kidney tissue extracted from D388-GGA and D388-WT infected birds, again confirming comparable *in vivo* replication kinetics. Of note, the presence of the marker silent mutations in nsp2 that differentiate D388-WT from D388-GGA were confirmed in IBV RNA isolated from tracheal tissue, ensuring that observed results were as a result of infection by the assembly-derived IBV. Together these data demonstrate that the Golden Gate Assembly-based reverse genetic system established in this study is a robust method for generating a clonal IBV which behaves similarly to wild type virus.

## Using the Golden Gate Assembly-based reverse genetics system to make single point mutations within the IBV genome

The most significant aspect of any reverse genetics systems is the ease with which modifications can be made to the genomic sequence. As a proof of concept, we generated two recombinant IBVs (rIBVs) using GGA that contained single point mutations in Nsp 10 and Nsp 14 (Fig 1B). The first rIBV, D388-nsp14mt-GGA contained a single mutation G18066C resulting in the amino acid change Val393Leu in Nsp 14. The second rIBV, contained the same mutation plus a single mutation C12089T resulting in the amino acid change Pro85Leu in Nsp 10. These two mutations were chosen as previous research has shown that incorporation of these mutations into the IBV strain M41 resulted in a temperature sensitive replication phenotype *in vitro* and an attenuated phenotype *in vivo* [53, 54]. These mutations offer a promising avenue for rational attenuation and therefore IBV vaccine development.

While it is possible to generate these point mutants *in silico* by simply ordering modified fragments D388-F6 and -F8, it can be cost prohibitive to order many 2kb fragments, particularly in cases where a large number of mutations may be of interest. Thus, for these point mutations C12089T and G18066C we demonstrate the ease of introducing mutations using a standard site directed mutagenesis protocol, modifying the plasmids containing fragments D388-F6 and -F8 respectively. The resultant plasmids were substituted into the GGA assembly protocol and two rIBVs were subsequently recovered in CK cells, D388-14mt-GGA and D388-10.14mt-GGA (S2 and S3 Tables). A stock of each rIBV was generated after two passages in embryonated hens' eggs and the sequence of the resulting rIBVs was confirmed to contain the desired modifications.

To determine whether either of the mutant rIBVs exhibited a temperature sensitive replication phenotype, a ciliary activity assay was carried out in *ex vivo* TOCs at both 37°C and 41°C. At 37°C ciliary activity in TOCs infected with D388-14mt-GGA (Fig 4B) and D388-10.14mt-GGA (Fig 4D) was comparable to those infected with D388-GGA suggesting comparable replication kinetics. At 41°C however ciliary activity was comparable to mock infected TOCs and was significantly higher than D388-GGA infected TOCs at all time points from 24 to 96 hpi (Fig 4B and 4D). This retention of ciliary activity indicates that both D388-14mt-GGA and D388-10.14mt-GGA are unable to replicate at 41°C; a finding that supports previous research using the M41 strain of IBV [54]. The replication of both mutation-containing rIBVs *in ovo* resulted in titres comparable to D388-GGA at 24 hpi, with all rIBVs exhibiting titres of over $10^6$ CD$_{50}$/ml (Fig 4A and 4C). Achieving high titres in embryonated hens' eggs is important as all current IBV vaccines are manufactured in this manner. The system presented here is therefore a powerful tool for rapidly creating mutant rIBVs to investigate pathogenic and immunogenic determinants within the IBV genome, and specifically those within a nephropathogenic IBV genome.

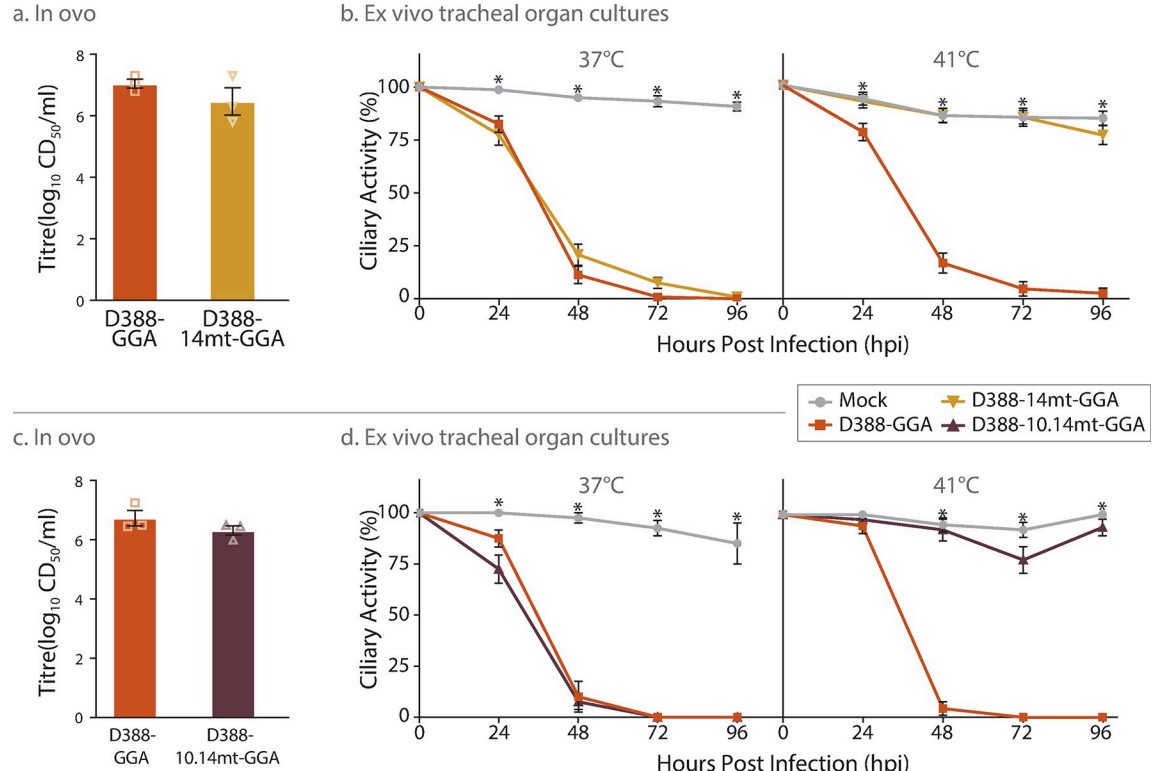

**Fig 4. Mutation Val393Leu in Nsp 14 confers a temperature sensitive replication phenotype.** (A and C) 10–11-day old SPF eggs were inoculated with $10^3$ $CD_{50}$ of either D388-GGA, D388-14mt-GGA, D388-10.14mt-GGA. The quantity of infectious progeny in the allantoic fluid 24 hpi was determined via titration. (B and D) In replicates of 10 *ex vivo* TOCs were inoculated with $10^3$ $CD_{50}$ of either D388-GGA, or D388-14mt-GGA, D388-10.14mt-GGA mock infected with media. Infected TOCs were incubated at either 37 or 41˚C and the ciliary activity observed at 24 h intervals. (A-D) Error bars represent SEM of three independent experiments. Statistical differences are highlighted by * (p<0.05) and were assessed (A, C) by a One way ANOVA and (B, D) a Two way ANOVA with a Tukey Test for post hoc analysis.

## Using Golden Gate based reverse genetics to make whole gene swaps

The S gene is important for pathogenicity, immunogenicity and tropism both *in vitro* and *in vivo*, not only for IBV but also other coronaviruses including porcine respiratory coronavirus (PRCV), Transmissible Gastroenteritis Virus (TGEV) and SARS-CoV-2 [2, 55–58]. The design of the IBV GGA-based reverse genetics system isolated the S gene on a single fragment, allowing for whole S gene swaps (Fig 1C and S2 Table). This is an important design functionality as the IBV S gene is a determinant of serotype and protectotype in which IBV strains are classified into groups which offer protection against specific IBV strains [59]. Previous research using the highly attenuated Beaudette strain of IBV has demonstrated that the Beaudette S gene allows for a unique ability to replicate in Vero cells, a cell line licensed for vaccine manufacturing [60]. Incorporation of the Beaudette S gene into M41 also resulted in attenuation *in vivo* with the resulting rIBV, M41K-Beau(S), eliciting 100% protection from virulent M41 challenge [61]. The S gene, and specifically the Beaudette S gene may therefore offer a mechanism for both rational attenuation and additionally a mechanism in which to move IBV vaccine manufacturing *in vitro*, removing the dependency on embryonated hens' eggs.

To demonstrate the ease in which the S gene could be swapped using the GGA reverse genetic system, the rIBV D388-Beau(S)-GGA was generated. In line with previous research, the ectodomain of the S gene from D388 was replaced with the equivalent sequence of rIBV

a. Chicken kidney cells

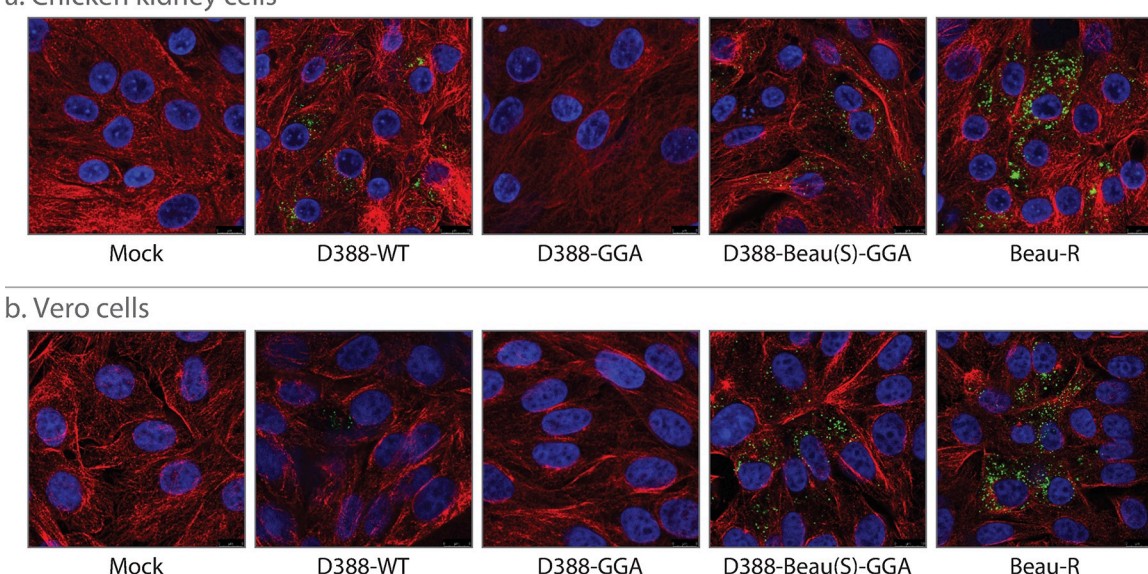

b. Vero cells

**Fig 5. The S gene from Beaudette confers the ability to replicate in Vero cells.** A Chicken Kidney cells and B Vero cells were mock infected or infected with MOI > 1 of D388-WT, D388-GGA, D388-Beau(S)-GGA or Beau-R. Infected cells were fixed 24hr post infection and immunolabelled with mouse double stranded RNA (dsRNA) and rabbit anti-tubulin. Primary antibodies were visualized using secondary antibodies, Alexa Fluor 488nm conjugated goat anti-mouse (green) and Alexa Fluor 568nm conjugated goat anti-rabbit (red). Nuclei were labelled with DAPI (blue).

Beau-R [61]. The rIBV Beau-R is a molecular clone of the attenuated laboratory strain, Beaudette-CK (Beau-CK) [36], which is in a different serotype, genotype and protectotype grouping to D388. To swap the D388 S gene for the Beau-R S gene, an alternative fragment D388-F10 was designed in which the ectodomain sequence of the D388 S gene was replaced with equivalent S sequence from Beau-R. This sequence was generated *in silico* and subsequently cloned into the holding vector. The assembly of the IBV genome was then carried out by substituting fragment D388-F10 for the alternative fragment D388-F10-Beau(S) containing the Beau-R S sequence. The rIBV D388-Beau(S)-GGA was recovered in CK cells, a stock was generated after three passages in CK cells, and the sequence of the altered S gene was confirmed. The ability to passage the rIBV during the recovery process in CK cells suggested that the incorporation of the Beaudette S had resulted in a tropism change. To confirm this and to determine whether D388-Beau(S)-GGA had the ability to replicate in Vero cells, both CK (Fig 5A) and Vero cells (Fig 5B) were infected with either D388-WT, D388-GGA, D388-Beau(S)-GGA, Beau-R [36], or mock infected. The cells were imaged using confocal microscopy using an antibody targeting double stranded RNA (dsRNA). As described earlier, during mock and D388-GGA infection no dsRNA could be detected in either CK or Vero cells. Conversely, in D388-Beau(S)-GGA and Beau-R infected cells, dsRNA was detected in both CK and Vero cells. This experiment demonstrates that the replacement of the wild type IBV D388 spike gene with Beau-CK S via Golden Gate Assembly confers the Beaudette strain's unique ability to replicate in Vero cells to the IBV strain D388, and shows the ease in which chimeric cDNA may be constructed using the one-pot GGA method.

## Discussion

The GGA-based reverse genetics system presented here is a simple, flexible method for investigating virus biology that will save critical time in avian IBV research. While prior vaccinia

virus-based reverse genetics systems for IBV involved several months of work and multiple rounds of passage through cells to create a single recombinant virus, typically taking upwards of 12 weeks [47], the GGA-based workflow presented here allows rescue of rIBV behaving similarly to wild type virus in mere days via direct rescue of assembled cDNA in primary cells. Since the emergence of SARS-CoV-2 the number of reverse genetics systems available for coronaviruses has increased and have largely replaced the well-established Vaccinia Virus based system [28, 62]. When designing a reverse genetics system or choosing which to use, a number of factors need to be considered including speed, cost, ease of sequence manipulation, sequence stability, method of viral recovery and laboratory facilities. For example, the yeast TAR based reverse genetics system, which has an excellent error and assembly rate comparable to Vaccinia Virus, takes roughly one to two weeks to assemble the full-length cDNA copy of the genome [63], but requires appropriate laboratory facilities for yeast propagation. The assembled construct can be used directly for the recovery of infectious virus or may require further amplification. Further amplification however adds the risk of incorporating sequence errors, particularly if using *E. coli* rather than yeast for this step. Assembly using BAC is not possible for all coronavirus genomes, with, for example, regions of IBV proving particularly toxic in *E. coli* [28]. Many of the systems require several steps for genome assembly. In contrast the protocol described in this study allows assembly to occur in a single step in a one-pot reaction. Furthermore, as demonstrated in this study, the assembly can be used directly for recovery of infectious virus, minimizing the risk of incorporation of sequence errors. The assembly protocol does not require specialized equipment or culturing facilities that are not available in a standard molecular biology laboratory.

While our GGA molecular clone does replicate comparably to the wild type virus in the natural host (Fig 3), it must be noted that some differences in replication were observed between the wild type virus and the GGA derived rIBV both *in vitro* and *ex vivo* (Fig 2). The ciliary activity assay in *ex vivo* TOCs (Fig 2B) suggests a lag in viral replication between the clone and the wild type, and additionally the clone appears to be unable to replicate in primary CK cells. While we cannot rule out that these phenotypic differences are a result of the introduced domesticating mutations (i.e. the silent mutations in nsp 2), the results in CK cells suggest an issue with cell entry (i.e. the S protein) rather than nsp 2. The replacement of the S gene with that derived from the laboratory strain Beau-R generating an rIBV able to infect both CK and Vero cells (Fig 5) supports this hypothesis. In further support, D388-GGA exhibited comparable replication to the wild type virus both *in ovo*, specifically at 24 hpi, and *in vivo* (Figs 2 and 3) suggesting that the marker mutations had not negatively impacted viral replication in the natural host. Sequencing of RNA recovered from *in vivo* infected tissues demonstrated the marker mutations had been retained, further suggesting that the marker mutations had not negatively impacted viral replication *in vivo*.

Wild type viruses exist as a quasi-species which consist of a large number of variant genomes, which individually may have different properties and phenotypes [64]. It is noteworthy that the consensus sequence used to generate D388-GGA was generated from D388-WT propagated in embryonated hens' eggs, where comparable replication between D388-WT and D388-GGA was observed, and not from tracheal cell culture or CK cells, where different phenotypes were observed. Differences in phenotypes between a molecular clone and the wild type virus have been observed previously with the inclusion of nucleotides from minor variants impacting the phenotype, either *in vivo* or *in vitro* of the molecular clone [30, 53, 65, 66]. Regardless, careful phenotyping of the base assembly is important for interpreting any downstream experiments, and, whether due to clonal genotype or specific domesticating mutations, it should not be assumed that synthetic viral genomes will behave identically to isolated WT. Despite the differences observed in specific virus-cell systems, the GGA-based reverse genetics

system described in this study is a powerful tool for the study of the D388 strain of IBV. Further, following establishment and characterization of the system, we demonstrate the utility of the GGA-based IBV reverse genetics system for studying point mutations and performing gene swaps requiring only simple molecular biology techniques.

Type IIS restriction enzymes have been previously used to assemble cDNA copies of coronavirus genomes, including SARS-CoV-2 and several other coronaviruses including IBV [67, 68]. In 2020, Xie *et al* divided the SARS-CoV-2 genome into 7 parts [33], which were cut using a mixture of Type IIS enzymes, gel purified, ligated, and used for *in vitro* transcription of SARS-CoV-2 RNA in a multi-step protocol, a similar methodology to Fang *et al* who assembled the cDNA genome of the Beaudette strain of IBV from 5 parts [67]. Additionally, a recent Golden Gate Assembly-based reverse genetics system was described for SARS-CoV-2 by Taha *et al*, which split the genome into 10 parts and then used a cycled assembly protocol followed by passage through *E. coli* using a BAC system [39]. While these previously published systems demonstrate the general utility of a genome assembly approach for studying coronavirus biology, the system presented here offers key improvements. In particular, the use of DAD to generate high fidelity assemblies by avoiding mismatch-prone ligation pairs is the key to high efficiency and high yield of final full-length, error-free product. This facilitates the direct use of assembly reactions in virus rescue, rather than passage through *E. coli* using a BAC system, which may be problematic for unstable genomes such as IBV. While it is possible the system described for the SARS-CoV-2 genome could be rescued directly, this was not demonstrated, and we note several predicted low-fidelity overhang pairs in that design [39], including the use of a palindromic overhang (TGCA, the F5-F6 fusion site) that would lead to a high level of misassembly. We would expect a larger proportion of misassembled product to reduce the amount of viable assembled genomes, which would in turn reduce the success of rescue experiments. Thus while GGA has now been shown to be a generally applicable method to generation of reverse genetics systems, careful design can ensure high reliability of the protocol to permit bypass of passage through *E. coli* cells and minimize sequence verification steps before rescue.

The method presented here reduces multiple steps of digestion, ligation, and passage through cells into a one-pot reaction that can be directly transfected for virus recovery, while incorporating features that promote flexible assembly design and biosafety measures. The cDNA fragments for assembly can either be ordered from a gene synthesis vendor or can be generated from genomic DNA samples using polymerase chain reaction (PCR) with the option of subsequently cloning into holding vectors to maximize DNA purity. As demonstrated through the generation of rIBVs which exhibit expected replication and pathogenic phenotypes, the flexible assembly design promotes simplified interrogation of specific mutations and allows for substitution of whole genes by swapping out component parts with minimal experimental effort. Additionally, it would be straightforward to convert the assembly into a replicon system for interrogation of specific steps in the viral life cycle, such as genomic RNA replication and subgenomic mRNA synthesis, recombination, assessment of antiviral compounds and the assessment of non-entry related host restriction factors [69–71]. A key factor in the generation of a replicon system is the ability to produce non-infectious viral particles, which for IBV is generally accomplished by eliminating the spike protein. With the described reverse genetics system here, this can easily be accomplished by replacing the S gene-containing fragment with one lacking the S gene. Propagation of the replicon cDNA can then be accomplished by insertion into a BAC or vaccinia virus. This inherent design flexibility provides the tools to study viral replication without producing infectious particles. This system therefore is not only conceptually simple and efficient, but also a powerful tool for IBV research that can used for the rapid response to emerging variants of IBV, which is particularly important to the development of vaccines to control spread within poultry populations.

We expect this method to be broadly applicable to eukaryotic viruses and especially useful for assembly of viruses with genomes too large to be contained on a single propagated plasmid ($> 10$kb). With the use of DAD, at least 30 fragments can be reliably assembled with high fidelity [44], and therefore even larger viral genomes than IBV are theoretically accessible using this method. We also note that the direct transfection of the assembled cDNA results in a clonal virus population, which could be essential for precise investigation of specific virus strains, as was most clearly and recently demonstrated during the SARS-CoV-2 pandemic [30, 72]. Ultimately, this precise, flexible, and simple reverse genetics system accelerates the time-scale of research on avian coronavirus to benefit both scientific discovery and the response to a disease of high concern.

## Methods

### Ethics statement

Primary cells and tracheal organ cultures (TOCs) prepared from chickens as well as all *in vivo* experiments were carried out in strict accordance with the Home Office guidelines of the United Kingdom (UK) and UK Animals (Scientific Procedures) Act 1986. Animals were raised and all procedures carried out in licensed facilities at The Pirbright Institute (TPI), establishment license number X24684464. Chickens for the *in vivo* experiment were purchased from The National Avian Research Facility (NARF) located at The Roslin Institute, UK, and delivered to TPI at 1 day of age. Embryonated hens' eggs for primary cells and organ cultures were also purchased from NARF. Embryonated hens' eggs for all other purposes including the propagation of viruses were purchased from VALO Biomedia, Germany.

The IBV reverse genetics system and all modifications to the IBV genome have been risk assessed and approved by The Pirbright Institute's Biological Agents and Genetic Modification Safety Committee (BAGMSC). All virus work was carried out in containment level 2 facilities at TPI.

### Cells and viruses

Primary chicken kidney (CK) cells were generated by trypsinization of kidneys harvested from 2–3-week-old specific pathogen free (SPF) Rhode Island Red (RIR) chickens by the cell culture department of TPI, following a previously published protocol [73]. TOCs were prepared from the same chickens as previously described [74]. All cell cultures unless otherwise stated in the method below were maintained at 5% $CO_2$, 37˚C and TOCs were maintained at 37˚C, 7–8 revolution per hour, no $CO_2$. Vero cells, a continuous cell line derived from the kidney of an African green monkey [75] were maintained in Eagle's minimum essential medium (EMEM) supplemented with 10% FCS at 37˚C, 5% $CO_2$.

All viruses were propagated in embryonated SPF RIR hens' eggs. Allantoic fluid was clarified by low-speed centrifugation. Viruses were either titrated in CK cells in triplicate or in *ex vivo* TOCs as described previously [76] and titers expressed as either plaque forming unit (PFU) per ml or as the dose required to cause 50% ciliostasis ($CD_{50}$) per ml. The nephropathogenic strain of IBV, D388 [8] was a gift from Professor J.J de Wit at The GD Animal Health, Netherlands; this virus is referred to here as D388-WT (wild type). Recombinant IBV (rIBV) Beau-R is a molecular clone of the Beau-CK strain, accession number AJ311317, and has been described previously [36].

### Next generation sequencing of IBV and rIBV

RNA was prepared from 2 ml of allantoic fluid that compromised the stock virus of D388-WT and the recombinant IBV (rIBV) D388-GGA following a previous published protocol [46]. As

previously published, library preparation for all samples was performed in triplicate with 100 ng total RNA for each sample, using a NEBNext directional Ultra II RNA-Seq kit (NEB, Ipswich, MA). Library QC was performed using the Bioanalyzer 2100 DNA 1000 kit and Qubit BR kit, prior to pooling. All libraries were sequenced on an Illumina MiSeq, using a MiSeq reagent v3 600 cycle cartridge. Analysis was performed using an in-house pipeline. Briefly, Trim Galore (http://www.bioinformatics.babraham.ac.uk/projects/trim_galore/) and fastqc (https://www.bioinformatics.babraham.ac.uk/projects/fastqc/) were used to perform initial read QC and quality filtering with minimum quality scores of 30 and minimum read length of 100 nucleotides. Host subtraction was performed by mapping all reads to the chicken genome (*Gallus gallus*) (GCA_000002315.5) using BWA-MEM [77]. All remaining reads were then subjected to a de novo assembly using SPAdes [78]. The resulting contigs were scaffolded using reference genome MN548289 [79] for D388-WT and the D388-WT sequence for D388-GGA. Reads were aligned back onto the *de novo* reference sequences and the resulting alignment was used to generate a consensus level reference sequence using BWA-MEM [77]. All nucleotide positions referenced in this manuscript relate to the deposited D388-WT sequence, GenBank accession number OR813926, unless otherwise stated.

## Determination of the genomic 5′ end sequence

RNA was extracted from 170 μL of the stock of D388-WT using a RNeasy mini kit (Qiagen) following the manufacturers method for RNA cleanup and the sequence of the 5′ UTR was determined using a 5′ RACE system for rapid amplification of cDNA ends (Invitrogen) using IBV-specific oligonucleotides 5′-TGTCTGCTCACTAAAC-3′ for the reverse transcription step and 5′-AGAACGTAGCCCAACGC-3′ for the amplification of dC-tailed cDNA step. Due to the large amounts of RNA structure in the 5′ UTR, the dT tailing reaction was performed on ice. The resulting PCR products were Sanger sequenced. The 5′ end sequence was determined to be ACTTAAGTGTGATATAAATATATATCATACATACTA.

## *In silico* avian infectious bronchitis virus cDNA assembly design

A Golden Gate Assembly system for wild type IBV strain D388 was designed using the suite of Data-Optimized Assembly Design tools available at https://ligasefidelity.neb.com. Assembly fusion sites were determined using the SplitSet tool and followed many of the optimization strategies described previously [44]. Briefly, the sequence of wild type IBV strain D388, GenBank accession number OR813926, was used as the core sequence. Silent mutations were required to remove two native BsaI sites in nsp2 (part of ORF1ab) and were selected using a codon usage table for *Gallus gallus* (T735A and T891A) (S1 Table).

Several features were added to the genome sequence as part of the assembly design. A T7 promoter was added upstream of the 5′ UTR (with RACE-determined 5′ end) followed by a GG dinucleotide to enhance transcription initiation. An encoded polyA tail ($A_{30}$) was added immediately after the 3' UTR. Following the polyA tail, a hepatitis delta virus (HDV) ribozyme sequence was included. A T7 terminator was added downstream of the HDV ribozyme sequence. The sequence was designed as a circular construct by the addition of a fusion site at the ends of these 5′ and 3′ extensions.

The SplitSet tool was used to divide this sequence into 12 parts via selection of 12 high fidelity fusion site sequences. The breakpoints were constrained to be between ORFs, with each fragment containing one or more complete coding sequences (Fig 1, S1 Fig and S2 Table). The set of fusion sites chosen have a fidelity score of 97% under the planned reaction conditions, indicating minimal possibility for mismatch ligation, thus maximizing potential yield of accurately assembled full-length product (S2 Fig).

All assembly parts (consisting of fragments F1 to F12 designed in the previous protocol with appended Type IIS cut sites) were ordered as sequence-verified plasmids in a pUC57--mini BsaI-Free backbone from Genscript at a maxiprep scale (S2 Table). An alternative fragment D388-F10-Beau(S), consisting of a chimera of the Beau-R spike protein, was also designed. D388-F10-Beau(S) had the nucleotides encoding the Spike sequence ectodomain (20,376 to 23,607) replaced with the corresponding sequence (20,418 to 23,640) from Beau-R (GenBank accession number AJ311317). Of note, the signal sequence (SS), transmembrane domain (TM) and cytoplasmic tail (CT) of the S sequence are retained from D388 in order to maintain interactions with the other D388 structural proteins. Additionally, a silent point mutation was made to the Beau-R S sequence (C20,907T, accession number AJ311317) to remove an internal BsaI restriction site.

## Generation of mutations within the IBV genome by Site-Directed Mutagenesis of fragments

The point mutation C12,089T was made to the plasmid encoding fragment D388-F6 resulting in the amino acid change Pro85Leu in Nsp 10 (nsp10.P85L) to create fragment D388-F6-10mt (S2 Table). The point mutation G18,066C was made to the plasmid encoding fragment D388-F8 resulting in amino acid change Val393Leu in Nsp 14 (nsp14.V393L) to create fragment D388-F8-14mt (S2 Table). Both point mutations were generated by site directed mutagenesis using the Q5 Site-Directed Mutagenesis Kit (NEB E0554S) using the wild type part-containing plasmids as the template and the following primers: Fragment D388-F6, forward 5′-CACCTGGGAAGTGCAG-3′ and reverse 5′-AGCTATATGTGCCCTGCAATAG-3′, Fragment D388-F8, forward 5′-TTCGTTACTTTGTAGGTATG-3′ and reverse 5′-TTTTCAGGATAACAATCCAC-3′.

## Assembly of IBV D388 cDNA

Four IBV cDNA constructs were generated by Golden Gate Assembly (S3 Table). D388-GGA contained the domesticated wild type IBV D388 sequence and the above described design features. Two rIBV cDNAs containing the described point mutations were generated by substituting mutation-containing fragments for the corresponding WT fragments: D388-14mt-GGA contained the nsp14.V393L mutation and D388-10.14mt-GGA contained both the nsp10.P85L and nsp14.V393L mutations. D388-Beau(S)-GGA, used the alternative fragment D388-F10-Beau(S) in place of D388-F10 to produce IBV D388 with a Beau-R S gene swap.

For all assemblies, each part was quantitated in triplicate using a Qubit assay (Thermo Fisher) and an equimolar fragment master mix was prepared to ensure a mole-balanced mixture of fragments. GGA reactions (20 μL final volume) were carried out with 3 nM of each DNA fragment from the master mix and 2 μL of the NEBridge Golden Gate Assembly Kit (BsaI-HFv2) (NEB E1601S) in 1X T4 DNA Ligase Reaction Buffer. Reactions were cycled between 37˚C and 16˚C for 5 min each for 90 cycles, followed by a final 60˚C heat soak for 5 min, before being stored at -20˚C until virus recovery experiments. 90 cycles were used to maximize yield of full length assembly product. Assembly reaction products were directly used in the following transfection and virus recovery protocols without amplification or purification. TapeStation (Agilent) analysis was optionally used to evaluate assemblies using the Genomic DNA ScreenTape following the manufacturer protocol (S3 Fig).

## Recovery of recombinant infectious virus in CK cells

The protocol for the recovery of rIBV from a Vaccinia Virus based reverse genetics was adapted from a previously published protocol [80]. Semi confluent CK cells seeded in 6 well

plates were infected with a recombinant Fowlpox virus that expresses a T7 RNA polymerase (rFPV-T7) [81]. Infected cells were then transfected using Lipofectin (Invitrogen) with 20 µl of assembled D388-GGA cDNA and 5 µg of a plasmid expressing the D388 N gene, also under the control of a T7 promoter. Transfection mix was incubated with the cells for 24 h at 37°C after which transfection mix was replaced with 3 ml BES [N,N-bis(2-hydroxyethyl)-2-ami-noethanesulfonic acid] medium. After a total of 48 h of incubation the supernatant was removed and discarded. Adherent cells were lysed in 1 ml BES media by one cycle of freeze-thawing (-80°C/37°C). The resulting cell lysate was filtered (0.22 µM) to remove rFPV-T7 and filtrate was inoculated into a 10 day embryonated SPF hens' egg (VALO). After 24 h, the embryos were culled by refrigeration for at least 4 h (a permitted method of humane culling) and the allantoic fluid harvested. The presence of infectious rIBV in the allantoic fluid was confirmed by a two-step reverse transcription polymerase chain reaction (RT-PCR) protocol using Protoscript II reverse transcriptase (NEB) and the random primer 5′-GTTTCCCAGT CACGATCNNNNNNNNNNNNNNNNN-3′ for the RT step and recombinant Taq polymerase (Invitrogen) for the PCR step using D388 specific primers 5′-GAGAGAAGGATTAGATTGTG-3′ and 5′-CCTGTATAATAGAAGTACCA-3′. Stocks of D388-GGA, D388-14mt-GGA, and D388-10.14mt-GGA were generated from two passages embryonated hens' eggs (VALO) and the quantity of infectious virus determined by titration in *ex vivo* TOCs [74]. Stocks of D388-Beau(S)-GGA were derived after three passages in primary CK cells. The sequence of the modifications in all rIBVs was confirmed by Sanger Sequencing.

## Assessment of ciliary activity in *ex vivo* tracheal organ cultures (TOCs)

TOCs were seeded one per tube and only those with 100% ciliary activity were used for the experiment. In replicates of at least ten, TOCs were inoculated with 1 ml TOC infection media (0.5 x EMEM, 75 mM a-methyl-D-glycoside, 40 mM HEPES, 0.1% sodium bicarbonate, 10 U/mL penicillin, 10 mg/mL streptomycin) containing $10^3$ $CD_{50}$ of either D388 (WT), D388-GGA, D388-14mt-GGA, D388-10.14mt-GGA, D388-Beau(S)-GGA or 1 ml media only for mock infection. TOCs were incubated at either 37 or 41°C. Using a light microscope, the ciliary activity of each TOC was assessed at 24 h intervals and graded as follows: 0, 25, 50, 75 or 100% activity as based on previously published protocols [48, 82].

## Assessment of replication *in ovo*

In replicates of three or five, 10–11 day-old SPF embryonated hens' eggs were inoculated with $10^3$ $CD_{50}$ IBV/rIBV and incubated at 37°C for 24 h. Eggs were chilled for at least 4 h at 4°C after which the allantoic fluid was harvested. The quantity of infectious progeny present in the allantoic fluid was determined via titration in *ex vivo* TOCs.

## *In vivo* assessment of viral replication

SPF RIR chickens were randomly assigned to one of three groups and housed in raised floor pens with enrichment including soft bedding, live feed and perches to promote natural animal behavior. Each group contained 20 chickens and was housed in separate positive pressure high efficiency particulate air (HEPA) filtered rooms; total number of animals in the experiment was 60. At 7 days of age each chicken was inoculated via the intra-nasal and -ocular route with 100 µl of PBS containing $10^{3.6}$ CD50 D388-WT, D388-GGA, or 100 µl PBS for mock infection. Chickens were assessed daily for the presence of IBV-induced clinical signs including snicking and rales, as described previously [53]. On 4- and 6-days post infection (dpi) five randomly chosen chickens per group were culled by cervical dislocation followed by decapitation, with blood and a variety of other tissues including trachea harvested. Harvested tissues were stored

in either PBS or RNAlater (Ambion, Thermo Fisher Scientific, Waltham, MA, USA) depending on the downstream application. All remaining birds were culled 7 dpi with tissues and blood collected. Ciliary activity was assessed in the harvested trachea on both 4 and 6 dpi. Throughout the experiment, which was a total of 8 days, all chickens were monitored at least twice daily. The humane endpoints were as follows: 1) sitting alone and not evading capture; the bird will be euthanised immediately, 2) respiratory distress e.g. excessive gasping; the bird will be euthanised immediately, 3) snicking and/or rales for seven days in total; the bird will be euthanised on the beginning to the 7th consecutive day and 4) excess drinking for more than two days as indicated by a fluid filled crop: the bird will be euthanised at the beginning of the 3rd consecutive day. None of the birds in the study died before meeting the criteria for euthanasia.

## Analysis of IBV-derived RNA in harvested tissues

Trachea samples stored in RNAlater were homogenized in 1 ml of Trizol using a Tissue Lyser II (30 Hz/s, 4 min). Homogenized samples were centrifuged to remove tissue debris. Per the manufacturer's (Invitrogen) protocol, 200 μl chloroform was added, samples centrifuged for 20 min at 12000 x g, 4˚C and the aqueous phase collected. RNA was precipitated using an equal volume of 70% ethanol after which, instead of a centrifuge step to pellet the RNA, the sample was loaded onto a RNeasy column (Qiagen). The remainder of the extraction followed the RNeasy tissue protocol (Qiagen) with an on-column DNase digestion step. The Luna Universal Probe One-Step RT-qPCR kit (NEB) following the manufacturer's protocol was used for the quantification of IBV derived RNA using IBV specific primers (forward 5′-GCTTTTGAG CCTAGCGTT-3′ and reverse 5′-GCCATGTTGTCACTGTCTATTG-3′) that target the 5′ UTR alongside the probe 5-6-carboxyfluorescein (FAM)-CACCACCAGAACCTGTCACCTC-6-carboxytetramethylrhodamine (TAMRA)-3′. Each reaction contained 500 ng of RNA.

## Analysis of IBV infection in CK and Vero cells by confocal microscopy

CK and Vero cells were seeded onto glass coverslips in 24 well tissue culture plates at $0.8 \times 10^6$ and $0.4 \times 10^5$ cells/ml, respectively, 3 days prior to infection. On the day of infection, cells were washed once in Phosphate-buffered saline a (PBSa) and inoculated with D388-WT, D388-GGA, D388-Beau(S)-GGA or Beau-R (a molecular clone of Beau-CK) [36] at an MOI of > 1 in a volume of 200 μl. BES medium was added to mock wells. Infected cells were incubated at 37˚C for 1 h (5% $CO_2$). After incubation 800 μl of BES medium was added to each well and cells were incubated for a further 23 h. Cells were washed once in PBSa, then fixed in PBSa containing 4% paraformaldehyde (Electron Microscopy Services, Hatfield, PA, USA) for 20 min at room temperature. Cells were washed once in PBS then permeabilized with PBS containing 0.1% Triton X100 (Sigma-Aldrich, St. Louis, MO, USA) for 10 min at room temperature. Cells were washed once in PBS, then blocking solution was added to each well consisting of PBS containing 0.5% bovine serum albumin (BSA, Sigma-Aldrich, St. Louis, MO, USA). Cells were incubated in blocking solution for 1 h at room temperature. Blocking solution was removed and replaced with blocking solution containing primary antibodies against dsRNA (Scicons, Szirák, Hungary) and alpha tubulin (Abcam, Cambridge, UK) (both diluted 1:1000). Cells were incubated with primary antibodies for 1 h at room temperature before three 5 min incubations in PBS. Secondary antibodies (AlexaFluor488 and 568 Goat Anti-Mouse, Invitrogen, Carlsbad, CA, USA) were diluted 1:400 in blocking solution and applied to cells for 1 h at room temperature (under foil). Cells were washed three times in PBS before the addition of 4′,6-diamidino-2-phenylindole (DAPI) (Abcam, Cambridge, UK) diluted 1:10,000 in $H_2O$. Cells were incubated with DAPI for 5 min before a final wash in $H_2O$. Coverslips were

mounted onto glass slides using VectaShield (Vector Labs, Burlingame, CA, USA) and sealed with nail varnish before analysis under a Leica confocal microscope.

## Statistics

All statistical analyses were completed using GraphPad Prism 9. Normality and standard deviation of each dataset was assessed before each statistical analysis. A one way ANOVA was used to analyze all *in ovo* experiments and a two-way ANOVA for ciliary activity and viral load in harvested *in vivo* tissues; post hoc analysis was a Tukey test. Statistical differences, $p < 0.05$, are highlighted by * in all figures.

## Supporting information

**S1 Fig. Schematic of the IBV D388-GGA construct noting the location of fusion sites and silent mutations used to remove native BsaI recognition sites.** A Genbank file of the full genome with the annotations is also provided as part of the Supporting Information.
(PDF)

**S2 Fig. Predicted assembly fidelity.** Data-optimized Assembly Design tools (ligasefidelity. neb.com) were used to check the fidelity of the assembly. The predicted ligation fidelity for the overhangs used in all IBV Golden Gate Assemblies (ACGG, ACAA, GTAA, CTCA, AACA, TAGG, CTGG, GGTG, TCCA, AGAT, CAAT, GAGC) is 97%. The reaction conditions used for this prediction were "BsaI-HFv2 37–16 cycling".
(PDF)

**S3 Fig. TapeStation gel to evaluate D388-GGA assembly reaction.** Assembly success was optionally assessed prior to rescue via visualization on TapeStation, a gel electrophoresis system that can visualize fragments up to ~40kb. The TapeStation molecular weight ladder is in lane 1. An example of pre- and post-assembly samples is shown in lanes 2 and 3, respectively. The assembly mastermix (lane 2) shows a range of sizes that reflects the 12 input vector parts. The shortest fragment (D388-F1) is 2451 bp and all other fragments range from 3741 bp to 5338 bp. Higher molecular weight species in lane 2 represent non-supercoiled populations of the input parts. Following the assembly reaction, the assembled D388-GGA cDNA (28 kb) (lane 3) runs between the two highest standards (15000 bp and 48500 bp). Under the cycling conditions used, the majority of the DNA is seen in this peak, with a second peak at 1.9 kb that is the empty vector backbone cut away from the fragments during assembly.
(DOCX)

**S1 Table. Domesticating mutations.** Silent mutations were made to remove native two BsaI sites from the IBV genome and Beau-R spike gene sequence.
(XLSX)

**S2 Table. Sequences of assembly fragments.** All fragments were held as plasmids in a pUC57-mini Bsa-Free backbone from Genscript. For each fragment, lowercase font indicates Type IIS sites and spacers and UPPERCASE indicates the insert sequence.
(XLSX)

**S3 Table. Component fragments for each rIBV assembly.**
(XLSX)

**S1 Raw images.**
(PDF)

**S1 File.**
(ZIP)

## Acknowledgments

We thank the central services unit, Animal Services department and Bioinformatics, Sequencing, and Proteomics Unit the at The Pirbright Institute for their help with preparation of primary chicken kidney cells, *the vivo* experiments and NGS respectively.

We thank Tom Evans, Siuhong Chan, Nathan Tanner, and James Eaglesham for helpful feedback during the preparation of this manuscript. We are grateful to Tasha Jose for assistance with figures.

## Author Contributions

**Conceptualization:** Sarah Keep, John M. Pryor, Gregory J. S. Lohman, Erica Bickerton.

**Data curation:** Graham Freimanis.

**Formal analysis:** Katharina Bilotti, Sarah Keep, Andrew P. Sikkema.

**Investigation:** Katharina Bilotti, Sarah Keep, Andrew P. Sikkema, John M. Pryor, James Kirk, Katalin Foldes, Nicole Doyle, Ge Wu, Graham Freimanis, Giulia Dowgier, Oluwapelumi Adeyemi, S. Kasra Tabatabaei.

**Methodology:** Katharina Bilotti, Sarah Keep, Andrew P. Sikkema, John M. Pryor.

**Supervision:** Sarah Keep, Gregory J. S. Lohman, Erica Bickerton.

**Writing – original draft:** Katharina Bilotti, Sarah Keep, Andrew P. Sikkema, Gregory J. S. Lohman, Erica Bickerton.

**Writing – review & editing:** Katharina Bilotti, Sarah Keep, Andrew P. Sikkema, Gregory J. S. Lohman, Erica Bickerton.

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
