## [Decision Letter · Decision Letter 0]

23 Apr 2024

PONE-D-24-07379One-pot Golden Gate Assembly of an avian infectious bronchitis virus reverse genetics systemPLOS ONE

Dear Dr. Bilotti,

Thank you for submitting your manuscript to PLOS ONE. After careful consideration, we feel that it has merit but does not fully meet PLOS ONE’s publication criteria as it currently stands. Therefore, we invite you to submit a revised version of the manuscript that addresses the points raised during the review process.

We look forward to receiving your revised manuscript.

Kind regards,

Haitham Mohamed Amer, PhD

Academic Editor

PLOS ONE

Journal Requirements:

"I have read the journal's policy and the authors of the manuscript have the following competing interests: 

When performing this research and drafting this manuscript, KB, APS, JMP, SKT, and GJSL were employees of New England Biolabs, a manufacturer and vendor of molecular biology reagents including DNA ligases and Type IIS restriction enzymes. New England Biolabs funded the work and paid the salaries of these authors. This does not alter our adherence to journal policies on sharing data and materials.

A patent has previously been filed by The Pirbright Institute to protect the intellectual property of the work surrounding the mutations in nsp 10 and nsp 14 (Patent name: Coronavirus, Number EP3172319B1, Authors: Erica Bickerton, Sarah Keep, and Paul Britton)."

Reviewers' comments:

Reviewer's Responses to Questions

**Comments to the Author**

1. Is the manuscript technically sound, and do the data support the conclusions?

Reviewer #1: No

Reviewer #2: Yes

Reviewer #3: Yes

2. Has the statistical analysis been performed appropriately and rigorously? 

Reviewer #1: No

Reviewer #2: Yes

Reviewer #3: Yes

3. Have the authors made all data underlying the findings in their manuscript fully available?

Reviewer #1: Yes

Reviewer #2: Yes

Reviewer #3: Yes

4. Is the manuscript presented in an intelligible fashion and written in standard English?

Reviewer #1: Yes

Reviewer #2: Yes

Reviewer #3: Yes

5. Review Comments to the Author

Reviewer #1: This research did not yield noteworthy findings regarding the current investigation into IBV. Instead, it presents a novel methodology better suited for journals emphasizing new protocols or methods. Notably, it highlights previously introduced point mutations using a distinct approach.

Introduction:

• The claim that "Coronaviruses are a family of viruses that infect a wide variety of species and that share a large (~30kb) positive sense RNA genome structure" is untrue. Coronaviruses mostly affect birds and mammals, not a large range of species. Furthermore, the size of the genome differs between coronaviruses, ranging from roughly 26 to 32 kilobases (kb), rather than a constant 30 kb.

• The introduction explains that "Avian infectious bronchitis virus (IBV), a gammacoronavirus and the first coronavirus identified, is the etiological agent of Infectious Bronchitis (IB)." Even though IBV is a gammacoronavirus, it wasn't the first one to be discovered. The mouse hepatitis virus was the first coronavirus to be identified in 1949. The human coronavirus OC43 was discovered in the 1960s.

• There should be more specificity added to the phrase, "Many IBV strains cause classical respiratory disease characterised by snicking, tracheal rales, watery eyes, nasal discharge, lethargy, reduced weight gain, and drops in egg production and quality."

• Regarding reverse genetics, the introduction mentions, "Classic reverse genetics systems for coronaviruses were first developed in the early 2000s based on RNA recombination in vivo, in vitro ligation using pre-existing or uniquely engineered restriction sites, or assembly and propagation of a cDNA copy of the coronavirus genome in a Bacterial Artificial Chromosome (BAC) or Vaccinia Virus Vector." While some reverse genetics systems were indeed established in the early 2000s, the field of reverse genetics for coronaviruses dates back further, with significant advancements made in the 1990s.

• The introduction talks about how a reverse genetics system for the betacoronavirus SARS-CoV-2 was developed using the Golden Gate Assembly (GGA). Nevertheless, the connection between this and the gammacoronavirus IBV and its reverse genetics system is not made clear.

Methods:

• The exact concentrations of reagents used in various steps, such as RNA extraction, Golden Gate Assembly reactions, and cell culture inoculation, are not provided.

• The sequencing procedure is not explained clearly enough. It does not specify which software tools or techniques were utilised for the assembly process, even though it claims using a "in-house pipeline" to put sequencing reads together.

• The name used to refer to IBV is inconsistent throughout the section; for example, it is called both "Infectious Bronchitis Virus" and "Infectious Bronchitis virus." It would be easier to read and understand if the terminology was standard.

Results:

• While the study describes the design of the reverse genetics system in detail, it lacks thorough validation of the design process. More information on the rationale behind specific design choices and experimental verification of the designed sequences would enhance the credibility of the approach.

• The introduction of silent mutations for marker purposes is a common approach; however, the study does not provide experimental evidence to confirm that these mutations do not affect viral replication or phenotype. Including functional assays to demonstrate the lack of impact of these mutations on viral fitness would strengthen the validity of the approach.

• While the study characterizes the replication kinetics of the recombinant viruses in vitro and in vivo, it lacks detailed analysis of key viral properties such as antigenicity, virulence, and host immune response. Additional experiments, such as serological assays or animal challenge studies, would provide a more comprehensive understanding of the behaviour of the recombinant viruses.

• The study could benefit from including more controls, such as viruses with unrelated mutations or wild-type viruses, to better interpret the observed phenotypic differences between the recombinant and wild-type viruses.

Reviewer #2: Katharina Bilott et al. present a novel one-pot reverse genetics system leveraging Golden Gate Assembly for the construction of the avian infectious bronchitis virus (IBV), a significant advancement in vaccine development. This methodology streamlines the generation of point mutants and gene replacements, offering a more efficient alternative to traditional methods. The IBV genome was segmented into 12 fragments, synthesized and amplified within E. coli plasmids, followed by in vitro assembly, showcasing the potential for creating point mutants and gene replacements. The manuscript is commendably structured, introducing an innovative and effective approach for IBV study and manipulation.

Specific points,

1. A comparison highlighting the advantages and disadvantages of this IBV construction method relative to other reverse genetics approaches would enrich the discussion. Such analysis could provide a clearer understanding of the method's uniqueness and practicality.

2. The efficiency of recovering recombinant IBV remains unclear. Additional details on the recovery rates and comparative efficiency with traditional methods could underscore the method's effectiveness.

3. On line 210, it is mentioned that "Next Generation Sequencing confirmed the sequence of the D388-GGA IBV genome". It would be beneficial to present these results, detailing the accuracy and fidelity of the assembled genome.

4. In line 544, the manuscript describes direct cell transfection with assembly reaction products without prior amplification or purification. Clarification is needed on how completeness and accuracy of the products are ensured without verification, potentially through product analysis or validation steps.

5. The manuscript would benefit from an assessment of the stability and concentration of D388. Exploring the challenges in assembling multiple fragments, especially identifying the most problematic aspects, would provide valuable insights into the method's robustness and limitations.

Reviewer #3: The study employed Golden Gate Assembly design and synthetic biology techniques to construct an efficient reverse genetics system for Infectious Bronchitis Virus (IBV). Demonstrating the system's application in generating point mutants and gene replacements, the team validated its feasibility and flexibility. These findings suggest that the reverse genetics system based on Golden Gate Assembly is a reliable and efficient tool for researching the biological characteristics of IBV and for vaccine development.

The results were well organized and presented, which is suitable for publication in this journal. However, some issues should be revised before publication.

Major:

1.The authors claimed no replication in the CK cells were observed by detecting the dsRNA with antibody, I wonder if this result has been confirmed by other available methods, eg: using specific antibodies against IBV N protein (which I think is more sensitive than detecting the dsRNA) or qPCR assay to detect sgmRNA at different time points.

2.The replication characteristics of the D388-WT and D388-GGA were compared in 10-day old embryo chicken eggs at only one time point-24 hpi, which can not fully demonstrate the replication characteristics, more time points should be added in this assay to draw an in ovo growth kinetics.

Minor:

1.L239: The specific infection dose of IBV in CK cells should be added as in 2A and 2B.

2.L241: The presentation of the phrase ‘mouse double stranded RNA (dsRNA) and rabbit anti-tubulin’ is incomplete, the canonical phrase should be ‘mouse double-stranded RNA (dsRNA) monoclonal antibody’.

3.L496: Is nsp2 a separate ORF?

4.L579: The unit was missing behind 103.

5.The page number were missing in more than one references.

6. PLOS authors have the option to publish the peer review history of their article (what does this mean?). If published, this will include your full peer review and any attached files.

Reviewer #1: **Yes: **Mohammed Rohaim

Reviewer #2: No

Reviewer #3: No

---

## [Author Response · Author response to Decision Letter 0]

14 Jun 2024

Dear Editors of PLOS ONE:

Thank you for reviewing our manuscript, “One-pot Golden Gate Assembly of an avian infectious bronchitis virus reverse-genetics system.” In this work, we describe the development of a robust, flexible, Golden Gate Assembly-based reverse-genetics system for infectious bronchitis virus (IBV), the infectious agent of infectious bronchitis, a disease of major veterinary concern for the poultry industry with significant implications for global food security. The speed and flexibility of our GGA-based reverse genetics system marks a significant development for both the study of IBV biology and vaccine development for this important pathogen. 

We greatly appreciate the insightful comments from the Reviewers, which contained several helpful suggestions to strengthen our work. We have submitted revised manuscript files with all edits highlighted as well as clean versions. Revisions to Figure legends are included in the Manuscript File below the paragraph in which the Figure is first cited. In addition, please see our point-by-point responses to address the Reviewers’ comments starting on the next page. The Reviewers’ comments are in black and our responses are in red. 

As requested by the Editors, we have also ensured that our manuscript meets the PLOS ONE style requirements. We have also updated the Competing Interests section to include the requested statement: 

When performing this research and drafting this manuscript, KB, APS, JMP, SKT, and GJSL were employees of New England Biolabs, a manufacturer and vendor of molecular biology reagents including DNA ligases and Type IIS restriction enzymes. New England Biolabs funded the work and paid the salaries of these authors. This does not alter our adherence to PLOS ONE policies on sharing data and materials.

A patent has previously been filed by The Pirbright Institute to protect the intellectual property of the work surrounding the mutations in nsp 10 and nsp 14 (Patent name: Coronavirus, Number EP3172319B1, Authors: Erica Bickerton, Sarah Keep, and Paul Britton). This does not alter our adherence to PLOS ONE policies on sharing data and materials.

Finally, as requested, we have provided the original uncropped image for Figure S3, a TapeStation gel which is the only gel image in this manuscript. The file is named “S1_raw_images”. If you have any further requests or questions please let us know.

We believe that the requested changes have strengthened the manuscript and we are confident that the work is now suitable for publication in PLOS ONE. Thank you for consideration of our revised manuscript, and we look forward to your decision.

Katharina Bilotti

Research Department

New England Biolabs

240 County Rd.

Ipswich, MA 01938

T: 978-998-8987

Email: kbilotti@neb.com

5. Review Comments to the Author

Reviewer #1: This research did not yield noteworthy findings regarding the current investigation into IBV. Instead, it presents a novel methodology better suited for journals emphasizing new protocols or methods. Notably, it highlights previously introduced point mutations using a distinct approach.

Response to reviewer: We respectfully disagree with the reviewer’s comment that our research has not yielded noteworthy findings. Our manuscript details the establishment of a one-pot reverse genetics system that permits the easy manipulation of the IBV genome on a greatly accelerated timeline compared to traditional methods. This presents a valuable tool for the continued study of this economically important pathogen. We further note that the PLOS ONE submission guidelines state the following: “PLOS ONE considers Research Article submissions which report new methods, software, databases and tools as the primary focus of the article. These should also adhere to the utility, availability and validation criteria in the guidelines for specific study types.” Upon reviewing the specific guidelines for a methods study in PLoS One, we strongly believe that our manuscript supports the utility, availability, and validation criteria as outlined.

Introduction:

• The claim that "Coronaviruses are a family of viruses that infect a wide variety of species and that share a large (~30kb) positive sense RNA genome structure" is untrue. Coronaviruses mostly affect birds and mammals, not a large range of species. Furthermore, the size of the genome differs between coronaviruses, ranging from roughly 26 to 32 kilobases (kb), rather than a constant 30 kb.

Response to reviewer: The text has been changed to reflect the reviewers comment that coronaviruses infect birds and mammals. The “~” symbol has been changed to the word “approximately.”

• The introduction explains that "Avian infectious bronchitis virus (IBV), a gammacoronavirus and the first coronavirus identified, is the etiological agent of Infectious Bronchitis (IB)." Even though IBV is a gammacoronavirus, it wasn't the first one to be discovered. The mouse hepatitis virus was the first coronavirus to be identified in 1949. The human coronavirus OC43 was discovered in the 1960s.

Response to reviewer: As indicated in the text, Reference 1 describes the discovery of the Beaudette strain of infectious bronchitis virus in 1937, which predates the discovery of mouse hepatitis virus in 1949. The infectious disease, Infectious Bronchitis (IB), was first described in 1931 (Schalk et al) with the cultivation of the associated infectious agent, IBV described in 1937. This is widely accepted in the field and additionally referenced in Nature Reviews Microbiology (V’kovski et al 2021) in which a timeline of notable events in the field is presented – both this reference and the reference describing IB in 1931 have been added to the statement. 

• There should be more specificity added to the phrase, "Many IBV strains cause classical respiratory disease characterised by snicking, tracheal rales, watery eyes, nasal discharge, lethargy, reduced weight gain, and drops in egg production and quality."

Response to reviewer: We are unsure of what additional specificity the reviewer would like to see included beyond the list of symptoms but would be happy to add additional references if any can be suggested.

• Regarding reverse genetics, the introduction mentions, "Classic reverse genetics systems for coronaviruses were first developed in the early 2000s based on RNA recombination in vivo, in vitro ligation using pre-existing or uniquely engineered restriction sites, or assembly and propagation of a cDNA copy of the coronavirus genome in a Bacterial Artificial Chromosome (BAC) or Vaccinia Virus Vector." While some reverse genetics systems were indeed established in the early 2000s, the field of reverse genetics for coronaviruses dates back further, with significant advancements made in the 1990s.

Response to reviewer: We apologize to the reviewer that our statement was incomplete and have added more detail. The first reverse genetics system that allowed the modification of a full-length cDNA copy of a coronavirus genome was published in 2000 by Almazan et al, and was based on transmissible gastroenteritis (TGEV). We have added this reference to the manuscript and clarified the text to read “Classic reverse genetics systems enabling the modification of a full-length cDNA copy of the coronavirus genome were first developed in the early 2000’s...” Prior to that the reviewer is correct in that advancements were made using RNA recombination; however, this method did not permit recovery of infectious clones from full length cDNA nor did it allow for the easy manipulation of Nsps, which account for two thirds of the genome. We have added the following to further acknowledge the advancements made in the 1990s - “The first method of modifying the coronavirus genome was described in the 1990s and was based on RNA recombination. It was originally based on Murine Hepatitis Virus (MHV) using temperature sensitive lesions within the N gene. While this was a significant advancement, the technology was limited in application and did not easily allow for modification of all areas of the genome; it was therefore was not widely adopted for all coronaviruses”. We have also added an appropriate reference to this section.

• The introduction talks about how a reverse genetics system for the betacoronavirus SARS-CoV-2 was developed using the Golden Gate Assembly (GGA). Nevertheless, the connection between this and the gammacoronavirus IBV and its reverse genetics system is not made clear.

Response to reviewer: We have updated this sentence:

“ Golden Gate Assembly (GGA) is widely used in synthetic biology to create systems for studying gene networks and metabolic pathways, and was recently applied to develop a new reverse genetics system for SARS-CoV-237.” 

to the following:

“ Golden Gate Assembly (GGA) is widely used in synthetic biology to create systems for studying gene networks and metabolic pathways, and was recently applied to develop a new reverse genetics system for another coronavirus, the betacoronavirus SARS-CoV-237.” 

Methods:

• The exact concentrations of reagents used in various steps, such as RNA extraction, Golden Gate Assembly reactions, and cell culture inoculation, are not provided.

Response to reviewer: We have added additional detail to the following sections of the Methods:

Determination of the genomic 5′ end sequence– We have added the quantity of virus and the RNA extraction kit used. The line now reads:

“RNA was extracted from 170 µl of the stock of D388-WT using a RNeasy mini kit (Qiagen) following the manufacturers method for RNA cleanup”

Analysis of IBV-derived RNA in harvested tissues- We have added the quantity of RNA added to the RT-qPCR reaction. The section now reads:

“The Luna Universal Probe One-Step RT-qPCR kit (NEB) following the manufacturer’s protocol was used for the quantification of IBV derived RNA using IBV specific primers (forward 5′-GCTTTTGAGCCTAGCGTT-3′ and reverse 5′-GCCATGTTGTCACTGTCTATTG-3′) that target the 5′ UTR alongside the probe 5-6-carboxyfluorescein (FAM)-CACCACCAGAACCTGTCACCTC-6-carboxytetramethylrhodamine (TAMRA)-3′. Each reaction contained 500 ng of RNA.”

We disagree with the reviewer regarding the detail for the Golden Gate assembly. The procedure is explained in full detail in the section “Assembly of IBV D388 cDNA” (see below): 

“For all assemblies, each part was quantitated in triplicate using a Qubit assay (Thermo Fisher) and an equimolar fragment master mix was prepared to ensure a mole-balanced mixture of fragments. GGA reactions (20 µL final volume) were carried out with 3 nM of each DNA fragment from the master mix and 2 µL of the NEBridge Golden Gate Assembly Kit (BsaI-HFv2) (NEB E1601S) in 1X T4 DNA Ligase Reaction Buffer. Reactions were cycled between 37°C and 16°C for 5 minutes each for 90 cycles, followed by a final 60°C heat soak for 5 min, before being stored at -20°C until virus recovery experiments. 90 cycles were used to maximize yield of full length assembly product.”

Similarly, we disagree about the methods for cell culture inoculation, which we presume means the quantity of virus used to infect cells or ex vivo TOCs. These methods are detailed in the following sections. For the response to reviewers comments we have underlined and put in bold the virus inoculation details. 

Assessment of ciliary activity in ex vivo tracheal organ cultures (TOCs).

TOCs were seeded one per tube and only those with 100% ciliary activity were used for the experiment. In replicates of at least ten, TOCs were inoculated with 1 ml TOC infection media (0.5 x EMEM, 75 mM a-methyl-D-glycoside, 40 mM HEPES, 0.1% sodium bicarbonate, 10 U/mL penicillin, 10 mg/mL streptomycin) containing 103 CD50 of either D388 (WT), D388-GGA, D388-14mt-GGA, D388-10.14mt-GGA, D388-Beau(S)-GGA or 1 ml media only for mock infection. TOCs were incubated at either 37 or 41°C. Using a light microscope, the ciliary activity of each TOC was assessed at 24 h intervals and graded as follows: 0, 25, 50, 75 or 100 % activity as based on previously published protocols48,82. 

Assessment of replication in ovo

In replicates of three or five, 10-11 day-old SPF embryonated hens’ eggs were inoculated with 103 CD50 IBV/rIBV and incubated at 37°C for 24 h. Eggs were chilled for at least 4 h at 4°C after which the allantoic fluid was harvested. The quantity of infectious progeny present in the allantoic fluid was determined via titration in ex vivo TOCs. 

Analysis of IBV Infection in CK and Vero Cells by Confocal Microscopy. 

CK and Vero cells were seeded onto glass coverslips in 24 well tissue culture plates at 0.8 x 106 and 0.4 x 105 cells/ml, respectively, 3 days prior to infection. On the day of infection, cells were washed once in Phosphate-buffered saline a (PBSa) and inoculated with D388-WT, D388-GGA, D388-Beau(S)-GGA or Beau-R (a molecular clone of Beau-CK)36 at an MOI of > 1 in a volume of 200 µl. 

• The sequencing procedure is not explained clearly enough. It does not specify which software tools or techniques were utilised for the assembly process, even though it claims using a "in-house pipeline" to put sequencing reads together.

Response to reviewer: The protocol used was identical to that already published in the indicated reference (Freimanis and Oade, 2020). We have however edited the section to include additional details of the pipeline. The section now reads: 

RNA was prepared from 2ml of allantoic fluid that compromised the stock virus of D388-WT and the recombinant IBV (rIBV) D388-GGA following a previous published protocol46. As previously published, library preparation for all samples was performed in triplicate with 100 ng total RNA for each sample, using a NEBNext directional Ultra II RNA-Seq kit (NEB, Ipswich, MA). Library QC was performed using the Bioanalyzer 2100 DNA 1000 kit and Qubit BR kit, prior to pooling. All libraries were sequenced on an Illumina MiSeq, using a MiSeq reagent v3 600 cycle cartridge. Analysis was performed using an in-house pipeline. Briefly, Trim Galore (http://www.bioinformatics.babraham.ac.uk/projects/trim_galore/) and fastqc (https://www.bioinformatics.babraham.ac.uk/projects/fastqc/) were used to perform initial read QC and quality filtering with minimum quality scores of 30 and minimum read length of 100 nucleotides. Host subtraction was performed by mapping all reads to the chicken genome (Gallus gallus) (GCA_000002315.5) using BWA-MEM77. All remaining reads were then subjected to a de novo assembly using SPAdes78. The resulting contigs were scaffolded using reference genome MN54828979 for D388-WT and the D388-WT sequence for D388-GGA. Reads were aligned back onto the de novo reference sequences and the resulting alignment was used to generate a consensus level reference sequence using BWA-MEM77. All nucleotide positions referenced in this manuscript relate to the deposited D388-WT sequence, GenBank accession number OR813926, unless otherwise stated. 

• The name used to refer to IBV is inconsistent throughout the section; for example, it is called both "Infectious Bronchitis Virus" and "Infectious Bronchitis virus." It would be easier to read and understand if the terminology was standard.

Response to reviewer: We fixed inconsistent capitalization throughout the text.

Results:

• While the study describes the design of the reverse genetics system in detail, it lacks thorough validation of the design process. More information on the rationale behind specific design choices and experimental verification of the designed sequences would enhance the credibility of the approach.

Response to reviewer: The design approach used here reflects our broader design approach, described in even more detail in the provided references (Potapov ACS Syn Bio 2018, Pryor PLOS ONE 2020, Pryor ACS Synthetic Biology 2022, Sikkema Current Protocols in Molecular Biology 2023). We have described the design for this specific assembly target in detail in the Results subsection “Design and Assembly of

---

## [Decision Letter · Decision Letter 1]

10 Jul 2024

One-pot Golden Gate Assembly of an avian infectious bronchitis virus reverse genetics system

PONE-D-24-07379R1

Dear Dr. Bilotti,

We’re pleased to inform you that your manuscript has been judged scientifically suitable for publication and will be formally accepted for publication once it meets all outstanding technical requirements.

Kind regards,

Haitham Mohamed Amer, PhD

Academic Editor

PLOS ONE

Reviewers' comments:

Reviewer's Responses to Questions

**Comments to the Author**

1. If the authors have adequately addressed your comments raised in a previous round of review and you feel that this manuscript is now acceptable for publication, you may indicate that here to bypass the “Comments to the Author” section, enter your conflict of interest statement in the “Confidential to Editor” section, and submit your "Accept" recommendation.

Reviewer #4: All comments have been addressed

Reviewer #5: All comments have been addressed

2. Is the manuscript technically sound, and do the data support the conclusions?

Reviewer #4: Yes

Reviewer #5: Yes

3. Has the statistical analysis been performed appropriately and rigorously? 

Reviewer #4: Yes

Reviewer #5: Yes

4. Have the authors made all data underlying the findings in their manuscript fully available?

Reviewer #4: Yes

Reviewer #5: Yes

5. Is the manuscript presented in an intelligible fashion and written in standard English?

Reviewer #4: Yes

Reviewer #5: Yes

6. Review Comments to the Author

Reviewer #4: The authors have addressed all my concerns. There response is sincere，I have no further comments for it.

Reviewer #5: Your manuscript titled 'One-pot Golden Gate Assembly of an avian infectious bronchitis virus reverse genetics system' describes an original methodology to create a rapidly practicable, reliable reverse genetics system for avian coronavirus infectious bronchitis virus. I hope your method described here will open an opportunity to vaccine researchers for quickly and custom tailored developing new generation recombinant or nucleic acid-based vaccines fully appropriate to newly emerging field IBV strains in the world.

7. PLOS authors have the option to publish the peer review history of their article (what does this mean?). If published, this will include your full peer review and any attached files.

Reviewer #4: **Yes: **Zhao Ye

Reviewer #5: **Yes: **Kamil Tayfun Carli

---

## [Editor Report · Acceptance letter]

17 Jul 2024

PONE-D-24-07379R1 

PLOS ONE

Dear Dr. Bilotti, 

I'm pleased to inform you that your manuscript has been deemed suitable for publication in PLOS ONE. Congratulations! Your manuscript is now being handed over to our production team.

Kind regards, 

on behalf of

Dr. Haitham Mohamed Amer 

Academic Editor

PLOS ONE